# GeSubNet: Gene Interaction Inference for Disease Subtype Network Generation

**Ziwei Yang[1,2], Zheng Chen[2]\*, Xin Liu[3]\*, Rikuto Kotoge[2], Peng Chen[4], Yasuko Matsubara[2], Yasushi Sakurai[2], Jimeng Sun[5]**

[1]Bioinformatics Center, Kyoto University, Japan    [2]SANKEN, Osaka University, Japan
[3]National Institute of Advanced Industrial Science and Technology (AIST), Japan
[4]RIKEN Center for Computational Science, Japan
[5]Department of Computer Science, University of Illinois Urbana-Champaign, USA
`yang.ziwei.37j@st.kyoto-u.ac.jp`
`{chenz, rikuto88, yasuko,yasushi}@sanken.osaka-u.ac.jp`
`xin.liu@aist.go.jp, peng.chen@riken.jp, jimeng@illinois.edu`

## Abstract

Retrieving gene functional networks from knowledge databases presents a challenge due to the mismatch between disease networks and subtype-specific variations. Current solutions, including statistical and deep learning methods, often fail to effectively integrate gene interaction knowledge from databases or explicitly learn subtype-specific interactions. To address this mismatch, we propose `GeSubNet`, which learns a unified representation capable of predicting gene interactions while distinguishing between different disease subtypes. Graphs generated by such representations can be considered subtype-specific networks. `GeSubNet` is a multi-step representation learning framework with three modules: First, a deep generative model learns distinct disease subtypes from patient gene expression profiles. Second, a graph neural network captures representations of prior gene networks from knowledge databases, ensuring accurate physical gene interactions. Finally, we integrate these two representations using an inference loss that leverages graph generation capabilities, *conditioned* on the patient separation loss, to refine subtype-specific information in the learned representation. `GeSubNet` consistently outperforms traditional methods, with average improvements of 30.6%, 21.0%, 20.1%, and 56.6% across four graph evaluation metrics, averaged over four cancer datasets. Particularly, we conduct a biological simulation experiment to assess how the behavior of selected genes from over 11,000 candidates affects subtypes or patient distributions. The results show that the generated network has the potential to identify subtype-specific genes with an 83% likelihood of impacting patient distribution shifts.

## 1 Introduction

Biological knowledge bases such as STRING (Szklarczyk et al., 2023) and KEGG (Kanehisa et al., 2024), and wet-lab experimental datasets such as gene expression data are crucial for understanding disease-gene association. While the knowledge bases are comprehensive, they often lack specificity for disease subtypes. This work introduces a deep learning method to integrate general knowledge bases with disease-subtype-specific experimental data to create more targeted knowledge graphs.

Decades of research have generated extensive disease-gene association data, compiled into various biological knowledge databases (Goh et al., 2007b; Szklarczyk et al., 2023; Kanehisa & Goto, 2000). These databases integrate known and predicted gene interactions, forming gene functional networks that describe how gene behaviors relate to disease processes. They support disease research by interpreting experimental results (Vella et al., 2017), facilitating biomarker discovery (Yang et al., 2023), and enabling personalized treatment (Goossens et al., 2015).

---

\*Corresponding authors.

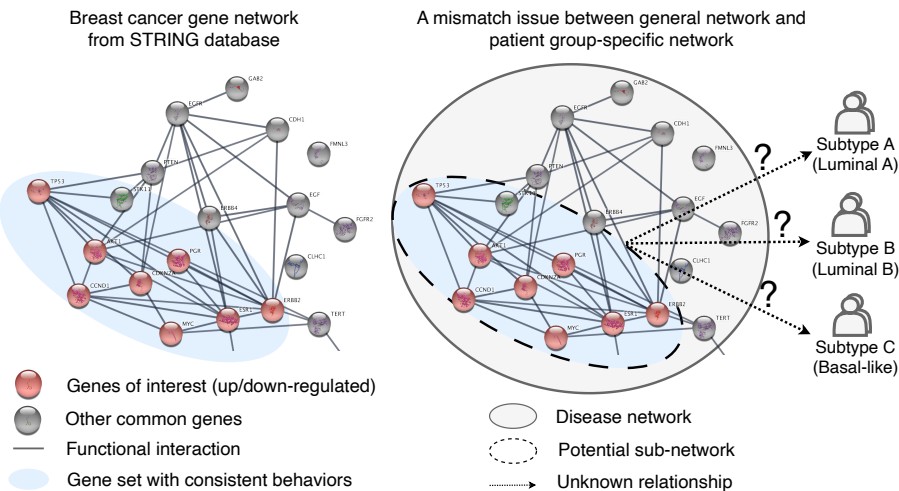

Figure 1: An example illustrating the mismatch issue in cancer gene networks. The BRCA gene network from the STRING database shows general interactions across various subtypes. Although a gene set with consistent behavior leads to the discovery of a sub-network, this sub-network cannot be directly linked to specific subtypes, such as Luminal A, Luminal B, or Basal-like.

Besides general knowledge bases, there are also in-lab experimental data, such as patient gene expression profiles. These experiments filter candidate genes, and the interactions in databases supported by these candidates are considered more relevant to subtypes. However, a mismatch exists between generic knowledge bases and experimental data when studying disease subtypes. For instance, as shown in Figure 1, breast cancer comprises multiple subtypes (luminal A, luminal B, and Basal-like), but databases like STRING provide only a general gene network for all subtypes. This generalization can lead to misinterpretations of gene behaviors across subtypes.

While bio-researchers have proposed data generation approaches to construct meaningful subtype-specific networks (Zaman et al., 2013), they often require extensive in-lab analyses such as pair-wise gene examination among hundreds to thousands of gene candidates. This paper introduces a novel data-driven approach to address this mismatch, automating the integration of gene expression data and knowledge databases to directly generate gene functional networks for various disease subtypes.

**Related Work.** Existing methods for generating subtype gene networks can be categorized into two groups: statistical and deep learning-based methods. Statistical methods focus on speeding up gene filtering by mining experimental data. These methods employ similarity metrics to measure the correlation between genes. High correlations, such as co-expressed genes (Zhang & Horvath, 2005), are marked as functional interactions. For example, ARACNe (Margolin et al., 2006) uses mutual information to measure expression similarity and removes indirect links with low similarity. WGCNA (Langfelder & Horvath, 2008) calculates Pearson correlation to support large-scale comparisons, while wTO (Gysi et al., 2018) transforms the correlations into probabilistic measures. However, gene interaction retrieval still prioritizes genes of interest.

A few deep learning methods leverage both knowledge databases and experimental datasets. They form disease networks as graphs and embed gene expression data, containing different patient information, as node embeddings. They set up link prediction and reconstruction using graph neural networks (GNNs). The newly reconstructed graphs can be viewed as specific networks. Representative methods include GAERF (Wu et al., 2021a), which learns node features with a graph auto-encoder and then uses a random forest to predict links. CSGNN (Zhao et al., 2021) predicts gene interactions using both a mix-hop aggregator and a self-supervised GNN. LR-GNN (Kang et al., 2022) proposes a dynamic graph method to gradually reconstruct graph structure, mitigating the constraints of prior general disease network information. Recent works focus on improving the accuracy of gene-gene link prediction (Li et al., 2024; Pang et al., 2024). However, their objective is only to reconstruct general disease-gene associations, including irrelevant interactions. This approach does not explicitly learn the distinct gene interactions unique to disease subtypes.

**Contributions and Novelty.** We present a new solution for leveraging distinct subtype information from experimental data, i.e., gene expression profiles, to directly infer **Ge**ne interactions specific to

disease **Sub**type **Net**works. This leads us to `GeSubNet`, which learns a unified representation that can accurately predict prior gene interactions while being able to distinguish different subtypes of a disease. Graphs generated by such representations can be considered subtype-specific networks. `GeSubNet` is a multi-step learning framework with independent data representation learning and integration. The first step uses a deep generative model to learn gene expression representations. These representations capture distinct data distributions and can distinguish subtypes in a latent feature space. The second step employs a GNN to learn graph representations of prior gene networks. This step ensures `GeSubNet` captures true gene-gene functional interactions collected in knowledge databases. Finally, we integrate the two representations, updating graph representations and inferring subtype-specific gene interactions using a reconstruction loss on the gene expression data. Our experiments confirm that `GeSubNet` can simultaneously generate different subtype networks within a general cancer. The contributions lie in:

- **Formulating New Gene Problem.** We first frame this problem as how to infer gene interactions can help models distinguish subtypes in experimental datasets. We investigate a method that automates the integration of gene expression data and knowledge databases, explicitly generating disease subtype networks.

- **Proposing Automated Data Integration Methodology.** `GeSubNet` is an effective architecture that combines a VQ-VAE and Neo-GNN, achieving average improvements of 30.6%, 21.0%, 20.1%, and 56.6% across three metrics on four cancer datasets. More advanced models can be easily integrated into `GeSubNet`.

- **Impacting Broad Biological Relevance.** We propose impactful biological evaluations and a new metric. The experiments involving 11,327 gene evaluations demonstrate that genes selected by `GeSubNet` are highly related to specific subtypes. We are the first to conduct a simulated experiment, termed Knock-out (Bergman & Siegal, 2003), to assess how the behavior of genes affects different subtypes. The proposed metric evaluates the reliability of selected gene interactions. The results show that `GeSubNet` effectively narrows down key genes.

- **Integrated Datasets for Cancer Subtyping.** We collect physical cancer-gene networks across four knowledge databases and construct machine-learning-ready datasets for experiments and evaluation. We release our datasets with this paper to support continued investigation. The code and data resources are available at: `https://github.com/chenzRG/GeSubNet`

## 2 PRELIMINARY AND PROBLEM SETTING

### 2.1 BACKGROUND: CANCER SUBTYPE

Cancer is a major public health concern with increasing incidence and leading to mortality. The National Cancer Institute (NCI) reports that the high costs of cancer care have been projected to grow to $246.6 billion by 2030 (COS, 2023). A key driver of these high costs and morbidity is cancer's inherent heterogeneity. Each cancer type is made up of multiple subtypes, characterized by distinct biochemical mechanisms, requiring specific therapeutic approaches (Balmain et al., 2003). While these subtypes may differ biochemically, they often share similar morphological traits, such as the physical structure and form of the organism (Yang et al., 2023), complicating precise diagnosis and treatment responses. This complexity highlights the need for deeper research into gene networks specific to cancer subtypes. However, as shown in Figure 1, current knowledge bases like STRING provide only broad cancer gene networks without distinguishing between subtypes. This limitation in specificity creates a gap in effectively targeting treatments based on unique subtype characteristics. Our paper addresses this problem by focusing on advancing research and tools that differentiate these subtypes at a more granular level.

### 2.2 PROBLEM SETTING

**Definition 1 (Gene expression data).** The fundamental entity in gene expression profile data is the individual patient. Each patient profile comprises tens of thousands of genes with measured features. Let $\mathbf{X} = \{\boldsymbol{x}^{(m)}\}_{m=1}^{M}$ denote a dataset of $M$ patients. Each patient can be represented as $N$ sequence of gene measures $\boldsymbol{x}^{(m)} = \{x_1^{(m)}, x_2^{(m)}, \cdots, x_N^{(m)}\}$. Let $\mathbf{Y} = \{y_1, y_2, \cdots, y_{|\mathbf{Y}|}\}$ denotes the set of subtypes for a cancer. Each $\boldsymbol{x}^{(m)}$ is associated with a label $y$.

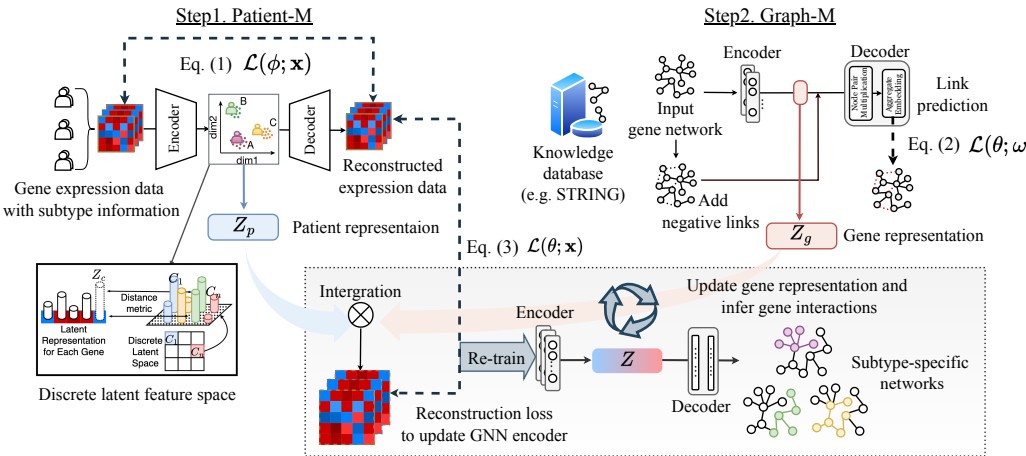

Figure 2: Overview: **Step 1: Patient-M** sets up an unsupervised learning task to generate the patient sample representation ($\mathbf{Z_P}$) from the input gene expression data ($\mathbf{X}$), which can distinguish subtypes. **Step 2: Graph-M** sets up a link prediction task to train the GNN encoder and decoder, learning the graph representation ($\mathbf{Z_g}$) from the input gene graph ($\mathcal{G}$) and expression data ($\mathbf{X}$). **Step 3: Infer-M** uses an objective function that integrates representations to generate subtype-specific networks. The reconstruction from Patient-M, conditioned on the GNN training in Graph-M ($q_\theta(\mathbf{z_g}|\mathcal{G})$), refines the graph structure, while ensuring accurate patient profile reconstruction ($p_\phi(\mathbf{x}|\tilde{\mathbf{x}})$).

**Definition 2 (Knowledge gene networks).** A gene network, as compiled in knowledge databases, can be represented as a general graph $\mathcal{G} = (\mathcal{V}, \mathcal{E})$ cross all $M$ patients, where $\mathcal{V}$ denotes the set of vertices, corresponding to the genes and $\mathcal{E}$ is the set of edges representing the gene interactions/links. Here, a link can be represented as $e_{ij} = (v_i, v_j)$, where $i, j \in N$.

**Problem (Subtype-specific gene network inference).** Given a general disease-gene network $\mathcal{G}$, we assume that it can be decomposed into a set of sub-graphs $\mathcal{G}_y = \{\mathcal{G}_1, \mathcal{G}_2, \ldots, \mathcal{G}_{|\mathbf{Y}|}\}$, corresponding to $\mathbf{Y}$ subtypes. The links, as defined in knowledge databases, are directly transformed into a set of edges $\{0,1\}^N \overset{\text{K.I.}}{\to} e_{ij} \in [0,1]^N$, where K.I. denotes knowledge-based initialization for graph construction. We aim to integrate the gene expression profile $\mathbf{X}$ to identify specific link sets relevant to a given subtype, formalized as $F(\cdot) : e_{ij} \to \{0,1\}^{(y)}$. Notably, these sub-graphs are not independent.

**Remark.** The function $F(\cdot)$ is designed by existing methods focusing on reconstructing the general graph $\mathcal{G}$. The learned representations only carry information for accurate reconstruction. In contrast, we investigate how to learn a representation from both data sources, one that captures essential information from gene interactions while distinguishing different subtypes. Our investigation is based on the following observation: the onset of complex diseases is typically attributed to changes (e.g., perturbations or disruptions) within a limited subset of genes (Goh et al., 2007a).

Formally, given $\mathbf{X}$ and a knowledge graph $\mathcal{G}$, we have $\{\mathcal{G}_1, \mathcal{G}_2, \ldots, \mathcal{G}_{|\mathbf{Y}|}\} = F(\mathbf{X}; \mathcal{G})$. We aim to learn a unified representation $\mathbf{Z}$ with two properties: (ⅰ) encode high-quality $\mathbf{Z}$ from gene expression profiles $\mathbf{X}$, that is, any $\boldsymbol{z}^{(m)}$ and $\boldsymbol{x}^{(m)}$ should correspond to the same patient group $y$; (ⅱ) enable $\mathbf{Z}$ to predict gene-gene interactions in $\mathcal{E}$. For the sub-graphs, we have two hypotheses:

- Hypothesis-1. The size of the sub-graph should be $|\mathcal{G}|_y \ll |\mathcal{G}|$, in terms of both the node set $\mathcal{V}$ and the link set $\mathcal{E}$, while having large margin differences with other sub-graphs.
- Hypothesis-2. $\mathcal{G}_y$ must maintain physical and biological meaningfulness. This is an important metric evaluated in experiments.

# 3 GESUBNET

## 3.1 FRAMEWORK

`GeSubNet` consists of three modules: patient sample representation learning module (Patient-M), graph representation learning module (Graph-M), and network inference module (Infer-M).

- **Patient-M**: This module sets up a cancer subtyping task, aiming to project patient gene expression profiles into a latent representation $\mathbf{Z_p}$, which can distinguish subtypes. This is typically an unsupervised learning task (Withnell et al., 2021; Yang et al., 2021b;a; 2023). `GeSubNet` employs a Vector Quantized-Variational AutoEncoder (VQ-VAE) (Van Den Oord et al., 2017) for two purposes: (i) to model this discriminative latent space using a flexible categorical distribution (Chen et al., 2023a), and (ii) to use the decoder as a key component of Infer-M.

- **Graph-M**: This module forms a link prediction task, leveraging both knowledge databases and gene expression data to learn $\mathbf{Z_g}$. The goal is to train a well-performed GNN autorencoder, where the encoder learns holistic gene interactions, and the decoder is used to generate new graphs. Since we focus on interactions, `GeSubNet` employs Neo-GNN (Yun et al., 2021), which combines structural information with node representations to prevent over-smoothing of node features.

- **Infer-M**: This module involves a novel objective function that integrates $\mathbf{Z_p}$ and $\mathbf{Z_g}$. `GeSubNet` uses the information from Patient-M, the decoder, and the reconstruction loss to optimize the prior knowledge in the gene network, i.e., the GNN encoder, for generating subtype-specific networks.

## 3.2 Subtype Gene Network Inference

**Gene Expression Representation Learning - Patient-M.** Given a gene expression dataset $\mathbf{X} \in \mathbb{R}^{M \times N}$, we first encode the gene expression profile to a low-dimensional embedding $\mathbf{Z}_e \in \mathbf{R}^{M \times D}$ through linear layers with ReLU activation function: $\mathbf{Z}_e = \text{ReLU}(\text{Linear}(\mathbf{X}))$, where $D$ is the dimension of $\mathbf{Z}_e$. We apply a Batch Normalization operation to prevent overfitting the limited patient gene expression samples. The $\mathbf{Z}_e$ is then projected along the $D$-axis into a set of feature vectors $\mathbf{Z}_c \in \mathbb{R}^{M \times D \times S}$, where $S$ denotes the vector dimension. Then, we project $\mathbf{Z}_c$ into a discrete codebook (Van Den Oord et al., 2017; Chen et al., 2023b). This involves encoding each dimension of gene features into a code, resulting in $\mathbf{Z}_p$ The codebook consists of $K$ latent vectors $\mathcal{P}_{1:K}$, which defines a $K$-way categorical distribution. The projection is conducted using the nearest neighbor search. Then, a decoder, consisting of linear layers with ReLU activations, reconstructs the original gene expression profiles, $\tilde{\mathbf{X}} \in \mathbb{R}^{M \times N}$.

**Gene Interactions Representation Learning - Graph-M.** Given general graphs $\mathcal{G}$ represented by an adjacency matrix $\mathbf{A}$ and gene expression data $\mathbf{X}$, we learn structural feature representations $\mathbf{X}' \in \mathbb{R}^{v \times u}$ using two MLPs: $\mathbf{X}' = \text{MLP}_{\text{node}}(\sum_{i=1}^{j} \text{MLP}_{\text{edge}}(A_{ij}), \mathbf{X})$, where the first MLP handles edges and the second handles nodes. Next, we encode $\mathbf{X}'$ with $\mathbf{A}$ to obtain graph representations $\mathbf{Z_g} \in \mathbb{R}^{N \times D}$. The GNN decoder computes similarity scores between paired node embeddings by first computing the element-wise product of $\mathbf{Z}_g^{(i)}$ and $\mathbf{Z}_g^{(j)}$. The resulting $D$-dimensional product is then aggregated into a single value as the similarity score. Finally, we train a binary classification MLP to perform the link prediction task: $\tilde{\mathbf{E}}_{ij} := \begin{cases} 1, & \text{Similarity Score} \geq 0.5 \\ 0, & \text{Otherwise} \end{cases}$ where 1 indicates the presence of a link between node $\mathbf{v}i$ and $\mathbf{v}j$, and 0 indicates the absence of a link. We use the predicted result $\tilde{\mathbf{E}}_{ij}$ to guide Graph-M in learning prior gene interaction knowledge.

**Subtype Network Inference - Infer-M.** This module integrates information from both Patient-M and Graph-M to optimize the prior cancer network and generate subtype-specific networks. We propose an objective function that uses Graph-M's graph generation capabilities, conditioned on the patient separation loss in Patient-M. `GeSubNet` follows three independent training phases.

Recall that we first train a well-initialized Patient-M to learn $\mathbf{Z_p}$ using gene expression profiles $\mathbf{X}$. This captures distinct subtype information through the following loss function:

$$\mathcal{L}(\phi; \mathbf{x}) := -\mathbb{E}_{q_\phi(\mathbf{z_e}|\mathbf{x})}[\log p_\phi(\mathbf{x}|\mathbf{z_q})] \tag{1}$$

where $\phi$ represents the parameters of the encoder and decoder. Then, we implement Graph-M to map predefined gene interactions for a given cancer into $\mathbf{Z_g}$:

$$\mathcal{L}(\theta; \omega) := -\frac{1}{E} \sum_{i=1}^{E} [h_e \log(\hat{h}_e(\omega; \mathbf{z}_e(\theta))) + (1 - h_e) \log(1 - \hat{h}_e(\omega; \mathbf{z}_e(\theta)))] \tag{2}$$

where $\theta$ and $\omega$ are the parameters of the encoder and decoder in the GNNs, and $h_e$ represents the ground truth for the presence of a gene interaction. After training $\mathcal{L}(\phi; \mathbf{x})$ and $\mathcal{L}(\theta; \omega)$, we first fix the model parameters $\phi$ and $\omega$, and reconstruct a new gene expression profile via matrix multiplication:

$\tilde{\mathbf{X}} = \mathbf{Z_p} \cdot \mathbf{Z_g}^T$. The reconstruction error between the integrated $\tilde{\mathbf{X}}$ and the original patient gene expression profile $\mathbf{X}$ is used to optimize the parameters $\theta$ of the graph encoder by:

$$\mathcal{L}(\theta; \mathbf{x}) = -\mathbb{E}_{q_\theta(\mathbf{z_g}|\mathcal{G})}[\log p_\phi(\mathbf{x}|\tilde{\mathbf{x}})] \qquad (3)$$

Here, the graph encoder conditions the reconstruction of patient or subtype-specific gene expression profiles. This ensures that graph representations capture the subtle characteristics of each patient's gene expression profile, inferring the newly generated links/interactions more relevant to subtypes.

## 4 EXPERIMENTS

### 4.1 DATASET AND PREPROCESSING

**Cancer gene expression dataset.** We collected the gene expression datasets from the world's largest cancer gene information database, The Cancer Genome Atlas (TCGA) (The Cancer Genome Atlas Research Network, 2013), across four cancer types: breast invasive carcinoma (BRCA) (Sharma et al., 2010), glioblastoma multiforme (GBM) (Urbańska et al., 2014), brain lower grade glioma (LGG) (Forst et al., 2014), and ovarian serous cystadenocarcinoma (OV) (Jayson et al., 2014). Detailed information can be found in Table 1 and Appendix B.1.
- *Preprocessing*: TCGA collected cancer samples from various experimental platforms with different patient information, such as gene sequencing results, and lacked alignments. First, we removed the unmatched gene IDs across cancer samples to ensure platform independence. Then, we identified and eliminated genes with zero expression (based on a threshold of more than 10% of samples) or missing values. Finally, we converted the scaled estimates in the original gene-level RSEM (RNA-Seq by expectation maximization) files to FPKM (fragments per kilo-million base) mapped reads data. The detailed data preprocessing pipeline can be found in Appendix B.2.

**Gene network dataset.** We collected gene functional networks corresponding to these four cancer types from four well-used knowledge databases, including KEGG (KE) (Kanehisa & Goto, 2000), STRING (ST) (Szklarczyk et al., 2015), InterPro (Int) (Paysan-Lafosse et al., 2023), and Monarch (Mona) (Mungall et al., 2017).

- *Preprocessing*: We searched and downloaded raw network data through website APIs. We mapped the gene IDs in the expression dataset to the standard format of Entrez Gene IDs (Maglott et al., 2010) in the networks. We stored gene interactions with the shared gene IDs across both datasets. Finally, we reconstructed the raw data as a binary matrix to initialize the gene graph construction. More details of the datasets and preprocessing can be found in Appendix B.3 and B.4.

Table 1: Summary of gene expression profile data and gene network data for four cancer types.

| Cancer | Gene Expression Matrix | | | Gene Network | | Knowledge Databases | | | |
|---|---|---|---|---|---|---|---|---|---|
| | #Subtypes | #Features | #Patients | #Nodes | #Edges | KE | ST | Int | Mona |
| BRCA | 5 | 11327 | 638 | 146 | 868 | ✓ | ✓ | ✓ | ✓ |
| GBM | 5 | 11273 | 416 | 102 | 203 | ✓ | ✓ | | ✓ |
| LGG | 3 | 11124 | 451 | 103 | 345 | ✓ | ✓ | ✓ | |
| OV | 4 | 11324 | 291 | 109 | 159 | ✓ | ✓ | | |

**Baselines.** We collected baselines from both the statistical methods and GNN-based methods. (1) The statistical methods include WGCNA (Langfelder & Horvath, 2008), which identifies modules of highly correlated genes using Pearson correlation; wTO (Gysi et al., 2018), which normalizes correlation by all other correlations and calculates probabilities for each edge in the network; ARACNe (Margolin et al., 2006), which calculates mutual information between pairs of nodes and removes indirect relationships; and LEAP (Specht & Li, 2017), which utilizes pseudotime ordering to infer directional relationships. (2) The GNN-based methods include GAERF (Wu et al., 2021a), which learns node features with a graph auto-encoder and a random forest classifier; LR-GNN (Kang et al., 2022), which generates node embeddings with a GCN encoder and applies the propagation rule to create links; and CSGNN (Zhao et al., 2021), which predicts node interactions using a mix-hop aggregator and a self-supervised GNN. More details are provided in Appendix C.

Table 2: Baseline comparison results on GED, DCS, and CDV for the proposed and baselines. GED, DCS, and CDV are subjected to min-max normalization. The best-performing results are highlighted in bold. The second-best results are highlighted in underline.

| Method | BRCA | | | GBM | | | LGG | | | OV | | |
|---|---|---|---|---|---|---|---|---|---|---|---|---|
| | CDV (↑) | GED (↑) | DCS (↓) | CDV (↑) | GED (↑) | DCS (↓) | CDV (↑) | GED (↑) | DCS (↓) | CDV (↑) | GED (↑) | DCS (↓) |
| WGCNA | 0.42 ± 0.02 | 0.39 ± 0.03 | 0.83 ± 0.04 | 0.43 ± 0.02 | 0.47 ± 0.03 | 0.83 ± 0.04 | 0.45 ± 0.03 | 0.53 ± 0.02 | 0.82 ± 0.04 | 0.24 ± 0.02 | 0.25 ± 0.03 | 0.83 ± 0.04 |
| wTO | 0.44 ± 0.02 | 0.43 ± 0.02 | 0.79 ± 0.03 | 0.45 ± 0.02 | 0.47 ± 0.02 | 0.83 ± 0.04 | 0.43 ± 0.03 | 0.59 ± 0.03 | 0.76 ± 0.04 | 0.26 ± 0.02 | 0.25 ± 0.03 | 0.83 ± 0.04 |
| ARACNe | 0.47 ± 0.02 | 0.45 ± 0.03 | 0.73 ± 0.03 | 0.44 ± 0.02 | 0.43 ± 0.02 | 0.79 ± 0.03 | 0.43 ± 0.03 | 0.57 ± 0.03 | 0.76 ± 0.04 | 0.23 ± 0.02 | 0.25 ± 0.03 | 0.81 ± 0.03 |
| LEAP | 0.49 ± 0.03 | 0.44 ± 0.03 | 0.78 ± 0.03 | 0.48 ± 0.03 | 0.45 ± 0.03 | 0.78 ± 0.03 | 0.44 ± 0.03 | 0.55 ± 0.02 | 0.77 ± 0.04 | 0.22 ± 0.02 | 0.24 ± 0.03 | 0.84 ± 0.04 |
| GAERF | 0.54 ± 0.06 | 0.58 ± 0.07 | 0.64 ± 0.05 | 0.46 ± 0.04 | 0.48 ± 0.06 | 0.76 ± 0.05 | 0.55 ± 0.05 | 0.56 ± 0.06 | 0.83 ± 0.07 | 0.34 ± 0.05 | 0.36 ± 0.06 | 0.82 ± 0.06 |
| LR-GNN | 0.54 ± 0.05 | 0.59 ± 0.06 | 0.62 ± 0.04 | 0.57 ± 0.06 | 0.61 ± 0.07 | 0.75 ± 0.05 | 0.56 ± 0.06 | 0.66 ± 0.07 | _0.72 ± 0.05_ | 0.34 ± 0.05 | _0.37 ± 0.06_ | 0.82 ± 0.07 |
| CSGNN | _0.65 ± 0.06_ | _0.66 ± 0.07_ | _0.52 ± 0.06_ | _0.65 ± 0.07_ | _0.64 ± 0.06_ | _0.74 ± 0.05_ | _0.58 ± 0.06_ | _0.68 ± 0.07_ | 0.73 ± 0.06 | _0.35 ± 0.05_ | 0.35 ± 0.06 | _0.80 ± 0.05_ |
| GeSubNet | **0.75 ± 0.04** | **0.78 ± 0.04** | **0.47 ± 0.05** | **0.73 ± 0.04** | **0.74 ± 0.05** | **0.67 ± 0.04** | **0.67 ± 0.05** | **0.74 ± 0.04** | **0.62 ± 0.05** | **0.45 ± 0.04** | **0.44 ± 0.04** | **0.75 ± 0.04** |

## 4.2 EXPERIMENT-I: NETWORK INFERENCE

**Objective.** This experiment evaluates the effectiveness of subtype-specific networks, following our Hypothesis-1: (1) $|\mathcal{G}|_y \ll |\mathcal{G}|$, ensuring the generated network is sparse compared to the original; (2) each subtype network exhibits structural differences from the others.

**Setup and Metrics.** We train `GeSubNet` for each cancer (the parameter settings can be found in Appendix D), and then evaluate the generated graphs for subtypes on two factors:

- Sparsity Assessment: we utilize the Coefficient of Degree Variation (CDV) (Pržulj, 2007) to measure the variability in gene nodes within a network. A higher CDV value indicates that most genes have very few interactions (edges). Thus, `GeSubNet` infers that the network becomes sparser because only a few active genes dominate the interactions in this subtype network.

- Graph Structural Differences: we employ the Graph Edit Distance (GED) (Gao et al., 2010) and the DeltCon Similarity (DCS) (Koutra et al., 2013) to measure structural differences in gene networks. GED captures local changes in gene interactions, while DCS evaluates global structural similarities. A high GED value indicates significant differences in gene interactions. Conversely, a high DCS implies high similarity.

**Results.** Table 2 presents `GeSubNet` significantly outperforms all baseline methods in terms of GED, DCS, and CDV metrics across four cancer types. Compared with the second-best baseline, CSGNN, `GeSubNet` achieves improvements of 35.8%/32.4%/20.2%/34.1% in terms of GED across all four tasks. Additionally, it delivers a relative reduction of 29.8%/13.5%/21.6%/19.3% in terms of DCS. For CDV, the improvements are 33.4%/13.7%/17.9%/15.3%, respectively. In summary, when evaluating BRCA, GBM, LGG, and OV, `GeSubNet` consistently achieves lower DCS scores and higher GED and CDV scores. This indicates that the generated subtype-specific gene networks are sparse but structurally unique, i.e., they are significantly different from each other. The OV results are apparently unsatisfactory, but this aligns with existing knowledge (Lawler et al., 2017) that OV is a challenging cancer type due to the limited available samples (only 291 patients in Table 1) and the lack of information on their pathogenic mechanisms in existing knowledge databases.

## 4.3 EXPERIMENT-II: BIOLOGICAL MEANINGFULNESS

**Objective.** While three graph metrics show the statistical significance of the generated network, this experiment further evaluates their biological relevance, following our Hypothesis-2. (1) Instead of structural differences, we further assess whether each network shows biologically functional differences from other networks. (2) We examine whether the generated networks have the potential to narrow down key genes that contribute more specifically to their respective subtypes.

**Setup and Metrics.** We hence conduct two experiments as follows:

- Gene Ontology (GO) Analyses (Ashburner et al., 2000a): This method counts the number of unique GO terms (under the category Biological Process (Desmedt et al., 2008)) associated with the genes in each network. GO terms describe gene functions across biological processes, molecular functions, and cellular components, enabling comparisons between gene networks. For example, if $GO(\mathcal{G}_1) := \{A, B, C\}$ and $GO(\mathcal{G}_2) := \{A, D, E\}$, where $\mathcal{G}_1$ and $\mathcal{G}_2$ represent two generated subtype gene networks. $GO(\cdot)$ denotes the sets of GO terms for two networks. Here the number of Enriched Biological Functions (#EBF) is 4, i.e., $\{B, C, D, E\}$, since $A$ is the shared GO term. We evaluate GO for each cancer dataset across all baselines. A high #EBF value indicates greater functional diversity and biological differences between subtypes.

- Simulated Gene Knockout (Bergman & Siegal, 2003): This is a computational technique that mimics the effects of gene knockout experiments without physically altering the genome. In this

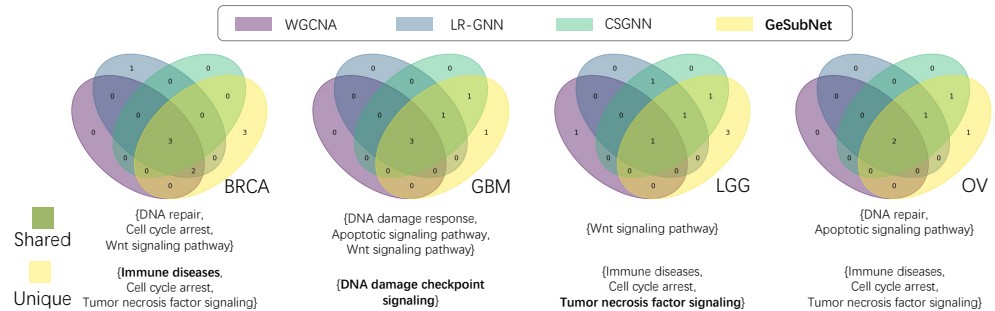

Figure 3: The Venn diagrams illustrate the overlap in GO terms resulting from different methods (WGCNA, CSGNN, LR-GNN, and `GeSubNet`) across four cancers. Shared and unique function items are listed here. A full list is provided in Appendix G. We highlight some unique function items that are well-supported by biological evidence in bold.

simulation, a gene is either deleted or deactivated to study its role within a specific subtype by observing changes in the patient sample distribution. As we described in an observation in Sec. 2, the key genes with significant expression differences form a small, limited set (Goh et al., 2007a), which leads to a distribution shift in patient samples during simulation experiments.

Our experiments follow three steps: (1) Rank all genes based on node degree disparities between the generated networks. (2) Group the genes into two sets: a high-ranking gene set and a low-ranking gene set, based on a threshold. (3) Individually simulate the knockout for high-ranking and low-ranking gene sets by transforming their expression values to a non-expression level.

To evaluate the results, this paper proposes a new metric Shift Rate ($\Delta_{\text{SR}}$) to measure the likelihood of distributional shifts in a subtype after a set of genes is knocked out. It calculates the average distance between the sample distributions before and after the knockout. We set a threshold ($\sigma_t$) based on the sample spread to assess the significance of distance. The $\Delta_{\text{SR}}$ is defined by :

$$\Delta_{\text{SR}} = \frac{1}{T} \sum_{t=1}^{n} \left( \frac{1}{n} \sum_{i=1}^{n} \|x_i^{\text{before}} - x_i^{\text{after}}\| > k \cdot \sigma_t \right) \tag{4}$$

where $T$ is the total number of knockout tests, $n$ is the number of patient samples within a subtype, $x_i$ represents an individual patient sample, $k$ is a scaling factor (e.g., 1.0 or 1.5) used to adjust the threshold, and $\sigma_j$ is the standard deviation of sample distances. Notably, this metric is only used after model training and cannot be involved in modeling training. More details on the simulated Gene Knockout can be found in Appendix H.

**Results.** Table 3 presents the results of the GO analysis , where our method consistently achieves the highest #EBF value across all datasets. These higher values indicate that the generated networks not only exhibit structural differences but also show functional distinctions from others from a biological perspective  (Ashburner et al., 2000b; Wu et al., 2021b).

Figure 3 presents Venn diagrams of detailed GO analysis for four cancer datasets, highlighting overlaps and unique in biological functions among three selected baselines and our method. We can consistently identify several unique functions across all datasets, while other methods rarely uncover unique functions, even when they achieve a comparable #EBF. For instance, in the LGG cancer dataset, CSGNN identifies 4 #EBF but finds no unique functions, whereas our method identifies 6 #EBF with 3 unique functions. From a biological perspective, our method demonstrates a robust array of enriched GO terms across different cancers, including pathways like Apoptotic signaling, Wnt signaling, Tumor necrosis factor signaling, and Cell proliferation. These terms represent critical cancer-related biological functions common to many cancers (Aktipis & Nesse, 2013), as shown in Table 9 in Appendix G. For

Table 3: The comparison results on #EBF between `GeSubNet` and the baselines. Only biological functions with high statistical significance (p-value $< 0.05$) are reported.

| Method | #EBF($\uparrow$) | | | |
|---|---|---|---|---|
| | BRCA | GBM | LGG | OV |
| WGCNA | 5 | 3 | 2 | 2 |
| wTO | 4 | 4 | 2 | 2 |
| ARACNe | 4 | 4 | 1 | 2 |
| LEAP | 3 | 3 | 2 | 3 |
| GAERF | 5 | 3 | 3 | 2 |
| LR-GNN | 6 | 4 | 3 | 3 |
| CSGNN | 3 | 5 | 4 | 4 |
| GeSubNet | **8** | **6** | **6** | **5** |

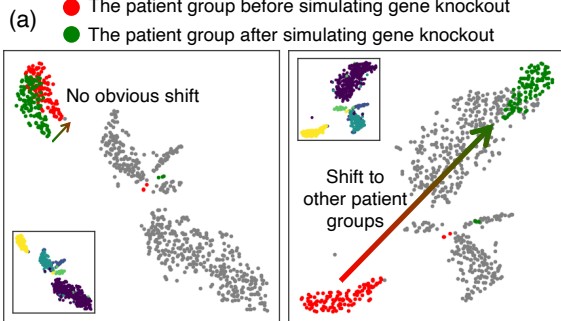

Table: Shift rates ($\Delta_{SR}$) on knocking out high- and low-ranking genes across different methods

| Method | Metric: $\Delta_{SR}$ | |
|---|---|---|
| | High-ranking genes ($\uparrow$) | Low-ranking genes ($\downarrow$) |
| WGCNA | 0.24 | 0.22 |
| wTO | 0.24 | 0.24 |
| ARACNe | 0.23 | 0.25 |
| LEAP | 0.28 | 0.24 |
| GAERF | 0.32 | 0.28 |
| LR-GNN | 0.23 | 0.23 |
| CSGNN | 0.34 | 0.29 |
| GeSubNet | **0.83** | **0.12** |

Figure 4: (a) UMAP visualization of an example showing patient distribution before and after the simulated gene knockout for a target subtype. The gray points in the main figure represent the negative control groups (subtypes). The small figures at the bottom left represent the original distributions of different subtypes. In the right subfigure, high-ranking genes are knocked out, while in the left, low-ranking genes are knocked out. (b) Table: shift rates ($\Delta_{SR}$) on knocking out high- and low-ranking genes, found by different baselines. The best results are highlighted in bold.

unique functions, our method identifies the "Immune diseases" function in BRCA, which has evident support as being related to breast cancer (McAlpine et al., 2012), and the "DNA damage checkpoint signaling" pathway, which is specific to GBM (Cheng et al., 2011).

Figure 4 shows the results of the simulated gene knockout experiments. The subfigure (a) visualizes an example of patient distribution before (red-marked points) and after (green-marked points) the Simulated Gene Knockout in both target and control groups (subtypes). In the left subfigure, there are almost no differences between the before and after distributions for the *low-ranking gene* set. In contrast, the right subfigure shows a significant shift in patient distribution, indicating that the suppression of high-ranking genes has a greater impact.

Figure 4(b) further provides a quantitative table across 11,327 genes in BRCA. our method achieves the highest $\Delta_{SR}$ for high-ranking genes, with an 83% likelihood of significantly shifting sample distributions. Meanwhile, the 12% $\Delta_{SR}$ for low-ranking genes suggests that our method effectively filters out common genes. Other baselines exhibit substantially lower $\Delta_{SR}$ values for high-ranking genes, ranging from 20%-30%, nearly matching those for low-ranking genes. Notably, while GNN-based methods like LR-GNN and CSGNN achieve comparable results in graph statistical metrics, their biological relevance is lower. This discrepancy arises because their objective functions aim only to reconstruct general disease networks, including irrelevant gene interactions, for all subtype samples. Although gene expression data embeddings result in different graph structures, these methods do not explicitly learn the distinct gene interactions unique to disease subtypes. However, learning a representation that incorporates prior knowledge while explicitly distinguishing patient samples in different subtypes is the key focus of this paper. This simulation experiment further validates the effectiveness of our method and demonstrates that `GeSubNet` maintains biological significance.

## 4.4 CASE STUDY

The case study on BRCA cancer follows established protocols in bioinformatics gene function studies (Huang et al., 2009). The analysis workflow is available in Appendix I.1 and I.2.
Figure 5 shows the gene networks A and B obtained for two BRCA subtypes. We observe that our method can generate gene networks with more distinct gene nodes. The networks show significant differences between the two subtypes, whereas the baselines produce more similar networks.
Figure 6(1) shows the gene expression distribution for the high-ranking and low-ranking gene sets. Different patient groups are marked in various colors to represent the ground truth. In the first column, we observe minimal differences in the expression distribution of low-ranking genes across patient groups. However, significant differences are evident in the high-ranking gene sets, as shown by the noticeable shift in distribution peaks.
Figure 6(2) presents expression heatmaps for the top three genes in both the high- and low-ranking gene sets. For the high-ranking set, the genes are ERBB2, CCNA2, and CCNE1, while the low-ranking set includes HHIP, MAPK1, and STK4. The high-ranking genes exhibit large differences in expression across subtypes, reflected by distinct color variations corresponding to the labels.

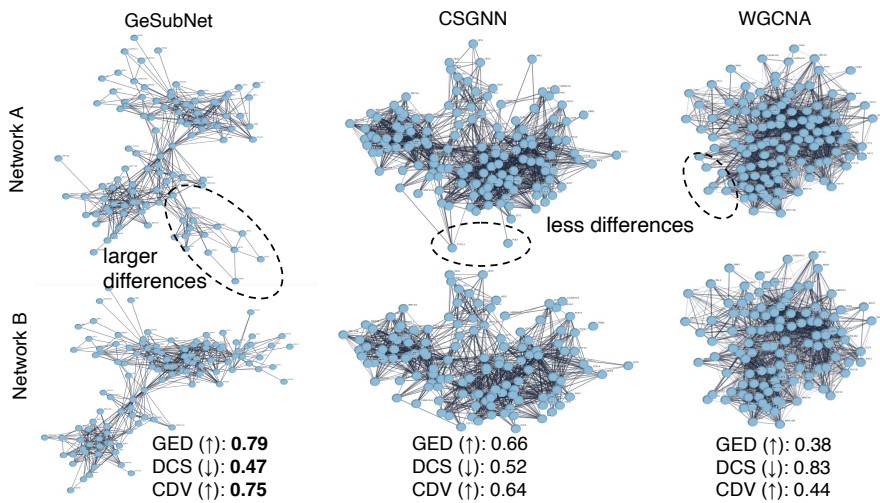

Figure 5: The obtained gene networks for two BRCA patient groups (subtypes).

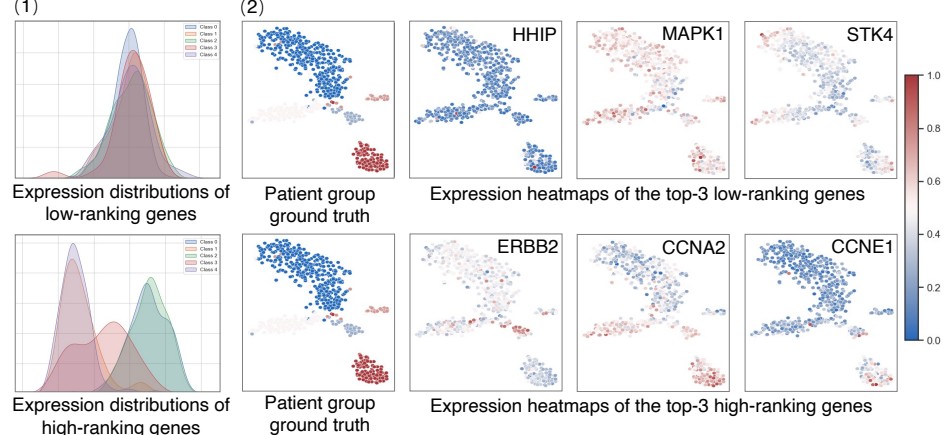

Figure 6: (1) expression level distributions and (2) the expression heatmaps of the top-3 genes from the high-ranking and low-ranking gene sets among different patient groups.

## 5 CONCLUSIONS

This paper introduced GeSubNet, a framework for inferring disease subtype-specific gene networks. GeSubNet includes sample and gene embedding learning modules that capture the characteristics of both patients and the prior gene graph. These embeddings are then utilized to reconstruct the input gene profile in the network inference module. This approach incorporates patient group information into the updated gene embeddings, enabling more accurate gene network inference specific to patient groups. In general, we explored a method for group-specific gene network inference on real-world clinical data. Importantly, we demonstrated the reliability of GeSubNet through a series of biological validations. We believe that continued investigation will bridge computational and biological sciences, advancing the understanding of diseases and foundational gene roles.

## 6 ACKNOWLEDGMENT

This work was supported by JSPS KAKENHI Grants-in-Aid for Scientific Research Number 24K20778 and 25K03231, JST CREST JPMJCR23M3, NSF award SCH-2205289, SCH-2014438, IIS-2034479, and AIST policy-based budget projects of "Research DX Platforms". This paper is based on results obtained from the project, "Research and Development Project of the Enhanced infrastructures for Post 5G Information and Communication Systems" (JPNP20017), commissioned by the New Energy and Industrial Technology Development Organization (NEDO).

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

# Appendix

CONTENTS

# A    NOTATIONS

All the mathematical notations and explanations used in the paper are summarized in Table 4.

Table 4: Mathematical notations and explanations.

| Notations | Explanations |
|---|---|
| $M$ | Number of patients |
| $N$ | Number of genes |
| $\mathbf{X}$ | Longitudinal gene expression data for patient $m$ |
| $\mathbf{Y}$ | Set of patient groups |
| $\mathcal{G}(\mathcal{V}, \mathcal{E})$ | Gene network represented as a graph |
| $\mathcal{V}$ | Vertex (or node) set representing genes in $\mathcal{G}$ |
| $\mathcal{E}$ | Set of edges representing associations between genes in $\mathcal{G}$ |
| $\mathcal{G}_y(\mathcal{V}_y, \mathcal{E}_y)$ | Sub-graph for patient group $y$ |
| $\mathcal{G}'$ | Reconstructed gene network |
| $F(\cdot)$ | Function to generate sub-graphs from edge information |
| $f_\theta$ | Function representing the model with parameters $\theta$ |
| $e_{ij}$ | Edge between gene $i$ and gene $j$ |
| $\mathbf{Z}, \mathbf{Z}_p, \mathbf{Z}_g$ | Lower-dimensional feature representation |

# B    DATASET

## B.1    GENE EXPRESSION DATA

*Gene expression* refers to the process by which information from a gene is used to synthesize functional gene products, typically *proteins*. This process is tightly regulated and varies between cell types, tissues, and environmental conditions, such as the tumor microenvironment (Brazma & Vilo, 2000). By measuring gene expression levels, researchers can determine the activity of specific genes within a cell or tissue at any given moment.

Gene expression data has a long history been used in cancer research (Zhang et al., 1997) because cancer is driven by the dysregulation of cellular processes, which often manifests in abnormal gene expression patterns. High-throughput technologies, such as RNA sequencing (RNA-Seq) and microarrays, gather patient gene expression profiles and simultaneously enable large-scale measurement of gene expression across thousands of genes (Liang & Pardee, 2003). Gene expression data allows researchers to study the molecular mechanisms hidden deeply in tumor development and progression.

The gene expression data used in this study were collected from The Cancer Genome Atlas (TCGA) (The Cancer Genome Atlas Research Network, 2013), obtained through the world's largest cancer gene information database Genomic Data Commons (GDC) portal (Grossman et al., 2016). All candidate patient samples were generated across various experimental platforms from cancer samples before treatment. For the cancer research community, it is common for available data to be contributed from various cancer study projects and institutions. As a result, the data are typically generated from different assay platforms. This non-uniformity of assay platforms introduces technical variations, such as differences in experimental protocols. These inherent batch effects pose a challenge as they can significantly impact downstream model training and any further analysis.

## B.2    PREPROCESSING OF GENE EXPRESSION DATA

To ensure platform independence, we initially removed the cross-platform lost genes. For the gene expression (transcriptomics) data generated from the Hi-Seq platform, we converted the scaled estimates in the original gene-level RSEM (RNA-Seq by expectation maximization) files to FPKM (fragments per kilo-million base) mapped reads data. We initially identified and removed all non-human expression features for the remaining data generated from the Illumina GA and Agilent array platforms. Subsequently, we applied a logarithmic transformation to the converted data. To eliminate potential noise, we identified and eliminated features with zero expression levels (based on a

threshold of more than 10% of samples) or missing values (designated as N/A). Table 5 describes the details of all experimental cancer gene expression datasets.

Preprocessing pipeline in **R (Ver.4.2.1)**:

**(1) Data Import:** Gene expression data were loaded after download.

```
data <- read.csv(``gene_expression_data.csv")
```

**(2) Filtering Low-Quality Samples:** Samples with a low number of expressed genes were removed using a default cutoff based on counts per million (CPM) values calculated with the edgeR package (Robinson et al., 2010).

```
keep <- rowSums(cpm(data) > 1) >= 10 filtered_data <- data[keep, ]
```

**(3) Normalization:** To account for differences in sequencing depth across samples, normalization was performed using the TMM (Trimmed Mean of M-values) method from the edgeR package.

```
norm_factors <- calcNormFactors(filtered_data)

normalized_data <- cpm(filtered_data, log=FALSE,
normalized.lib.sizes=TRUE)
```

**(4) Batch Effect Correction:** To minimize batch effects arising from non-uniform experimental protocols, the 'ComBat' function from the SVA package (Leek et al., 2012) was applied to remove unwanted variation across different platforms and projects.

```
corrected_data <- ComBat(dat=normalized_data, batch=batch_info)
```

**(5) Log Transformation:** The gene expression data were log-transformed to stabilize variance across genes.

```
log_data <- log2(normalized_data + 1)
```

**(6) Missing Data Imputation:** Missing expression values were imputed using the 'impute' function from the impute package (T et al., 2022).

```
imputed_data <- impute.knn(log_data)$data
```

### B.3 GENE NETWORK DATA

To obtain refined and coherent prior gene networks, we curated a comprehensive dataset by amalgamating information from diverse sources, including KEGG (Kanehisa & Goto, 2000), STRING (Szklarczyk et al., 2015), InterPro (Paysan-Lafosse et al., 2023), and Monarch (Mungall et al., 2017). These repositories collectively provide information on a broad spectrum of gene interaction corroborated by evidence from high-throughput lab experiments, co-expression analyses, genomic context predictions, disease-related gene pathways, and previously published knowledge.

### B.4 PREPROCESSING OF GENE NETWORK DATA

Our detailed preprocessing follows: We initiated the network construction process by retrieving related gene information through database APIs for a specified target cancer entry available in the databases above. To ensure uniformity in gene identifiers across disparate datasets, we harmonized gene IDs to the standard format of Entrez Gene IDs (Maglott et al., 2010). Subsequently, we identified and included common genes across all database sources as candidate nodes for constructing the prior network. Next, we retained common gene-gene associations obtained from multiple databases for each candidate node pair as the final edges to be preserved. Concurrently, isolated nodes were systematically removed from the network. During this curation of edges, we implemented two distinct screening strategies to elucidate two types of networks with edges embodying distinct correlation properties: (1) we identified edges denoting that the proteins are integral components of a physical complex, denoted as edges of Type *I*; and (2) we retained edges indicative of functional and physical protein associations, denoted as edges of Type *II*. This approach enhances our prior gene network by capturing diverse aspects of gene relationships and interactions. Table 6 describes the details of all experimental cancer gene network datasets. KEGG, STRING, InterPro, and Monarch are abbreviated as KE, ST, Int, and Mona, respectively.

Table 5: Descriptions of four cancer gene expression datasets.

| Cancer | Raw Transcriptomics | | | Gene Expression Matrix | |
|--------|-------|---------|--------|-----------------------|--------------|
|        | #Gene | #Patient | #Group | Sample size | Feature size |
| BRCA | 19537 | 638 | 5 | {320, 124, 119, 54, 21} | 11327 |
| GBM  | 17455 | 416 | 5 | {125, 111, 80, 68, 32} | 11273 |
| LGG  | 16245 | 451 | 3 | {213, 151, 87} | 11124 |
| OV   | 17226 | 291 | 4 | {81, 76, 68, 66} | 11324 |

Table 6: Descriptions of the four cancer gene network datasets.

| Cancer | Data Source | | | | #Node | #Edge (Type I) | #Edge (Type II) |
|--------|----|----|-----|------|-------|----------------|-----------------|
|        | KE | ST | Int | Mona |       |                |                 |
| BRCA | ✓ | ✓ | ✓ | ✓ | 146 | 289 | 579 |
| GBM  | ✓ | ✓ |   | ✓ | 102 | 75  | 128 |
| LGG  | ✓ | ✓ | ✓ |   | 103 | 206 | 139 |
| OV   | ✓ | ✓ |   |   | 109 | 46  | 95  |

## C    BASELINES

### C.1    THE PRINCIPLE OF SELECTING BASELINE METHODS

We collected baselines from both statistical methods and GNN-based methods. For statistical methods that do not involve neural networks, we first included two of the most classical and widely used methods in the bio-network computing field: **WGCNA** (Langfelder & Horvath, 2008) and **ARACNe** (Margolin et al., 2006). These methods have continually contributed to bio-network studies for many years.

In addition, we want to include more recently proposed methods in this category. Our selection principles were as follows: (1) the method was recently published (and approximately a decade after the above classical methods), (2) it provides open-source code (available on platforms such as GitHub), (3) it is easy to use (with outputs compatible with popular analysis packages), and (4) it has been used as a baseline in related bio-network research. Based on these criteria, we selected two methods, **wTO** (Gysi et al., 2018) and **LEAP** (Specht & Li, 2017).

For the GNN-based methods, our principles aligned with the above five lines for selecting newly proposed statistical methods. We selected three methods (i.e., **GAERF** (Wu et al., 2021a), **LR-GNN** (Kang et al., 2022), and **CSGNN** (Zhao et al., 2021)) that meet these principles.

### C.2    BASELINE METHODS

**(1) Weighted Gene Co-expression Network Analysis (WGCNA)** (Langfelder & Horvath, 2008) utilizes Pearson correlation to identify modules of highly correlated genes, where genes within the same module are likely to be functionally related or involved in similar biological processes. **(2) Weighted Topological Overlap (wTO)** (Gysi et al., 2018) normalizes the chosen correlation by all other correlations and calculates a probability for each edge in the network. **(3) Algorithm for the Reconstruction of Accurate Cellular Networks (ARACNe)** (Margolin et al., 2006) calculates the mutual information between pairs of nodes and then removes indirect relationships during network building. **(4) Lag-based Expression Association for Pseudotime-series (LEAP)** (Specht & Li, 2017) utilizes pseudotime ordering to infer the directionality between genes in the network. **(5) Graph Auto-encoder and Random Forest (GAERF)** (Wu et al., 2021a) learns features of nodes by a graph auto-encoder and concatenates features of two nodes as input for the random forest classifier. **(6) Link Representation-Graph Neural Network (LR-GNN)** (Kang et al., 2022) generates embeddings using a GCN encoder and then applies a propagation rule to create link representations for predicting associations in networks. **(7) Contrastive Self-supervised Graph Neural Network**

Table 7: Hyperparameter sensitivity experiment. The best-performing results are highlighted in bold, and the checkmark indicates our choice of the optimal settings.

| Hyperparameters etrics | GED ($\uparrow$) | DCS ($\downarrow$) | CDV ($\uparrow$) |
|---|---|---|---|
| Latent Dim = 16 | 0.76 | 0.49 | 0.74 |
| Latent Dim = 32 ($\checkmark$) | 0.78 | **0.47** | **0.75** |
| Latent Dim = 64 | **0.79** | 0.48 | 0.73 |
| #Code Book = 16 | 0.72 | 0.51 | 0.68 |
| #Code Book = 32 ($\checkmark$) | **0.78** | **0.47** | **0.75** |
| #Code Book = 64 | 0.75 | 0.54 | 0.63 |
| Batch Size = 16 | 0.76 | 0.48 | 0.73 |
| Batch Size = 32 ($\checkmark$) | **0.78** | **0.47** | **0.75** |
| Batch Size = 64 | 0.77 | 0.49 | 0.68 |

**(CSGNN)** (Zhao et al., 2021) predicts node interactions in networks by employing a mix-hop aggregator and a contrastive self-supervised GNN. WGCNA, wTO, ARACNe, and LEAP are well-used traditional methods that use only non-graph gene expression data as input, while GAERF, LR-GNN, and CSGNN are deep learning-based methods that use known paired integrations or networks as input. These methods are reported to have competitive performance for similar tasks like molecular interaction prediction. It is also worth noting that these methods tend to perform better when supplementary data, such as sequence data, is available.

## D  HYPERPARAMETER SETTING

We conducted parameter sensitivity experiments to determine the optimal hyperparameters. The results are presented in Table 7. Overall, the findings indicate that the model's performance is not significantly affected by changes in the hyperparameters.

## E  COMPUTATIONAL REQUIREMENTS

Table 8: The computational requirements of the proposed method, including both runtime (training and inference time) and memory usage, across different cancer datasets.

| Dataset | Training Time (sec) | Inference Time (sec) | GPU Memory Usage (MB) |
|---|---|---|---|
| BRCA | $6.23 \pm 0.03$ | $2.21 \pm 0.05$ | $4393 \pm 152$ |
| GBM | $3.64 \pm 0.04$ | $1.44 \pm 0.07$ | $2764 \pm 117$ |
| LGG | $5.96 \pm 0.03$ | $1.98 \pm 0.04$ | $3834 \pm 122$ |
| OV | $3.45 \pm 0.04$ | $1.25 \pm 0.04$ | $2583 \pm 108$ |
| Pan-cancer | $454.47 \pm 12.2$ | $142.86 \pm 5.2$ | $15893 \pm 733$ |

We evaluated the computational requirements of the proposed method, including runtime (training and inference time) and memory usage, across four cancer datasets. While cancer patient gene expression datasets typically contain hundreds of patients for each cancer type, we assessed the method's scalability by conducting an additional test using the TCGA Pan-Cancer expression dataset based on the BRCA graph. This dataset includes 8,314 patient samples and is one of the largest pan-cancer datasets. The results are summarized in the Table 8. We find that the computational requirements of the proposed method are manageable for real-world cancer datasets and scale effectively to larger datasets with more patient samples.

## F  EVALUATION METRICS

**(1) Graph Edit Distance (GED)**. GED (Gao et al., 2010) measures dissimilarity between graphs by quantifying the minimum cost required to transform one graph into another through a series of edit operations, such as adding or deleting nodes and edges and modifying node or edge attributes. GED

between two gene networks $N_1$ and $N_2$ is defined as: $GED(N_1, N_2) = \min_\pi \sum_{(u,v) \in \pi} c(u, v)$. Here $\pi$ is a set of edit operations, typically represented as a set of pairs $(u, v)$ where $u$ is a node in $N_1$ and $v$ is a node in $N_2$. This set $\pi$ represents the optimal alignment or correspondence between nodes in the two networks. $c(u, v)$ is the cost associated with aligning nodes $u$ and $v$, depending on factors such as node attributes, edge attributes, or the type of edit operation. We calculate the overall GED among $n$ inferenced networks as: $GED(N_1, N_2, \ldots, N_n) = \frac{1}{n(n-1)} \sum_{i=1}^{n} \sum_{j=1, j \neq i}^{n} GED(N_i, N_j)$.

**(2) DeltCon Similarity (DCS)**. It is a similarity score calculated through the DeltCon algorithm (Koutra et al., 2013). DCS quantifies the structural similarity between two graphs by comparing the influence of nodes across these graphs. It relies on the computation of the influence matrix derived from the graph Laplacian. The similarity is based on how the node influences the change in values between the two graphs. DCS is mathematically defined as: $DCS(G_1, G_2) = 1 - \frac{1}{2} \sum_{i=1}^{N} \sum_{j=1}^{N} \left( \sqrt{\frac{1}{N} \sum_{i=1}^{N} (\mathcal{I}_{G_1}(i, j) - \mathcal{I}_{G_2}(i, j))^2} \right)$, where $N$ is the number of nodes in the graphs, and $\mathcal{I}_{G_1}(i, j)$ and $\mathcal{I}_{G_2}(i, j)$ represent the influence of node $i$ on node $j$ in graphs $G_1$ and $G_2$, respectively.

**(3) Coefficient of Degree Variation (CDV)**. The degree distribution of a gene network represents the frequency distribution of node degrees, indicating the number of interactions with each gene. In other words, this variation in connectivity suggests that the network's degree distribution implies that certain genes are more central or connected than others, and these central genes may have crucial roles in defining or influencing specific cancer subtypes. CDV (Pržulj, 2007) also decreases as the average degree ($\bar{k}$) of the network increases. CDV is defined as: $CDV = \frac{\sqrt{\frac{1}{N} \sum_{i=1}^{N} (k_i - \bar{k})^2}}{\bar{k}\sqrt{N}} \times \frac{1}{\bar{k}}$. Here, $N$ is the total number of nodes, $k_i$ is the degree of node $i$, and $\bar{k}$ is the average degree.

**(4) Number of enriched biological functions (#EBF)**. Corresponding to the differences in graph properties of gene networks, we also explore their biological significance. A commonly used method for this is functional enrichment analysis, which identifies biological functions, pathways, and molecular activities that are overrepresented within a gene set when compared to a random selection of genes with a similar size and degree distribution from the genome. In our study, we performed Gene Ontology (GO) enrichment analysis using the R package `clusterProfiler` (Ashburner et al., 2000a), which leverages data from databases such as KEGG and GO to identify enriched biological terms. A greater degree of enrichment suggests that the network exhibits more meaningful gene interactions than would be expected by chance. This unique enrichment across subtypes implies that the gene networks represent biologically significant interactions, where genes within specific cancer subtype networks are functionally connected as a group. To evaluate the functional diversity between two gene networks, we conducted an experiment using GO to count the number of unique GO terms associated with the genes in each network. Specifically, we used the `enrichGO()` function from `clusterProfiler` to map the genes from both networks to their corresponding GO terms. The `compareCluster()` function was applied to compare the sets of GO terms associated with each network and to identify differences, focusing on the number of enriched biological functions. To quantify the differences, we calculated the number of enriched biological functions (#EBF) using the symmetric difference between the sets of GO terms. Mathematically, this is represented as: $\#EBF = (GO(\mathcal{G}_1) \setminus GO(\mathcal{G}_2)) \cup (GO(\mathcal{G}_2) \setminus GO(\mathcal{G}_1))$. This operation captures the unique functions present in one network but not another. Enrichment was evaluated based on statistical significance, where the biological functions with a p-value $< 0.05$ were reported. A higher #EBF indicates that the networks capture different biological processes or molecular functions, potentially reflecting the underlying biological differences between the networks' contexts.

## G GO FUNCTION ENRICHMENT ANALYSIS

### G.1 GO (GENE ONTOLOGY) AND BIOLOGICAL PROCESS

Gene Ontology (GO) (Ashburner et al., 2000a) is a comprehensive and standardized framework for annotating genes based on their functions in biological contexts. It provides a structured vocabulary to describe the roles of genes in three broad categories:

- **Biological Process (BP)**: This category defines the biological objectives or events that the gene products are involved in. BP terms focus on cellular functions that go awry in diseases like cancer. It includes essential processes such as "signal transduction", "cell cycle regulation", "immune response", and "metabolic processes".

- **Cellular Component (CC)**: This category defines the cellular locations where the gene products carry out their functions. CC terms focus on the spatial aspect of gene activity. Examples include terms like "cytosol", "nucleus", "mitochondrion", and "plasma membrane".

- **Molecular Function (MF)**: This category outlines the biochemical activities of the gene product. MF terms focus on how genes contribute to cellular machinery on a molecular level. Examples include terms like "ATP binding", "enzyme activity", and "protein binding".

The **Biological Process (BP)** category is of particular relevance to cancer research because it directly reflects the underlying mechanisms that drive cancer development and progression and, therefore, serves as a suitable representative class for cancer-related GO terms (Desmedt et al., 2008). Identifying disruptions in BP terms related to cancer mechanisms can also help guide therapeutic strategies. For instance, drugs targeting cell cycle checkpoints, such as cyclin-dependent kinase inhibitors, or those promoting apoptosis, like Bcl-2 family inhibitors, are being developed to specifically correct or mitigate BP-related dysfunctions (Shapiro, 2006; Ashkenazi et al., 2017).

## G.2  GO Function Enrichment Analysis Results

Table 9 presents the enriched Gene Ontology (GO) terms associated with various biological functions across four cancer types (BRCA, GBM, LGG, and OV), as identified using different methods in the GO analysis of Experiment II. The Venn diagrams in Figure 3 illustrate the overlaps among the results from the different methods. Due to the complexity of comparing multiple methods, we present a four-way Venn diagram focusing on four selected methods (WGCNA, CSGNN, LR-GNN, and GeSubNet) for clarity.

`GeSubNet` **findings:** `GeSubNet` shows a robust array of enriched GO terms across different cancers, including:

- **Apoptotic signaling pathway:** A series of biochemical events leading to programmed cell death, which is essential for eliminating damaged or unwanted cells and maintaining tissue homeostasis. Dysregulation of apoptosis is a hallmark of cancer.

- **Wnt signaling pathway:** A network of proteins involved in cell signaling that regulates important processes such as cell proliferation, migration, and differentiation. Aberrant Wnt signaling is often implicated in cancer development.

- **Tumor necrosis factor signaling:** A signaling pathway that can induce inflammation, apoptosis, or cell survival, depending on the context. It is involved in various aspects of cancer biology, including tumorigenesis and immune response.

- **Cell proliferation:** The process by which cells divide and multiply, essential for growth and tissue repair. In cancer, deregulated cell proliferation leads to tumor growth and cancer progression.

This set of terms encompasses a range of crucial cancer-related biological functions shared by most cancers. This indicates that the resulting gene network maintains physical and biological meaningfulness, i.e., the backbone consists of genes involving the main cancer progression.

**Comparison:** The proposed method identifies a broader range of distinct GO terms compared to other methods, and the GO term set identified by `GeSubNet` constitutes a superset of the terms determined by different methods.

For instance, in BRCA, WGCNA and CSGNN identify terms primarily focusing on cell cycle regulation, DNA repair, and apoptosis. wTO and ARACNe report similar functionalities with notable overlaps. GAERF and LR-GNN overlap more with the proposed method but still do not capture as many terms as the proposed method. The proposed method's overlap with other approaches is significant, particularly regarding core cancer pathways such as DNA repair (present in all methods), Cell cycle arrest (common in most methods), and Apoptotic signaling pathways (reported by several methods). However, the proposed method finds unique terms, such as immune diseases in BRCA,

Table 9: Detailed enriched GO terms across four cancer tasks resulting from different methods.

| Method | BRCA | GBM | LGG | OV |
|---|---|---|---|---|
| WGCNA | Cell cycle arrest, DNA repair, Apoptotic signaling pathway, Regulation of cell migration, Wnt signaling pathway, DNA repair | Cell cycle arrest, DNA damage response, Apoptotic signaling pathway | Wnt signaling pathway, Regulation of cell migration | DNA repair, Apoptotic signaling pathway |
| wTO | Cell cycle arrest, Apoptotic signaling pathway, Regulation of cell migration, DNA repair | DNA damage response, Apoptotic signaling pathway, Tumor necrosis factor signaling, Wnt signaling pathway, DNA damage response | Wnt signaling pathway, DNA repair | Regulation of cell migration, DNA repair |
| ARACNe | Cell cycle arrest, Apoptotic signaling pathway, Regulation of cell migration | Tumor necrosis factor signaling, Wnt signaling pathway, Cell proliferation | Wnt signaling pathway | DNA repair, Apoptotic signaling pathway |
| LEAP | Cell cycle arrest, DNA repair, Wnt signaling pathway, DNA repair | DNA damage response, Apoptotic signaling pathway, Wnt signaling pathway | Regulation of cell migration, Wnt signaling pathway | DNA repair, Apoptotic signaling pathway, Cell proliferation |
| GAERF | Cell cycle arrest, Wnt signaling pathway, Apoptotic signaling pathway, Regulation of cell migration, DNA repair | DNA damage response, Wnt signaling pathway, Cell proliferation | Cell cycle arrest, Wnt signaling pathway, Regulation of cell migration | DNA repair, Apoptotic signaling pathway |
| LR-GNN | Cell cycle arrest, Wnt signaling pathway, Apoptotic signaling pathway, Regulation of cell migration, DNA damage response | Wnt signaling pathway, DNA damage response, Apoptotic signaling pathway, Cell proliferation | Cell cycle arrest, Wnt signaling pathway, DNA repair | DNA repair, Apoptotic signaling pathway, Cell proliferation |
| CSGNN | DNA repair, Cell cycle arrest, Apoptotic signaling pathway | DNA damage response, Apoptotic signaling pathway, Wnt signaling pathway, Cell proliferation, Tumor necrosis factor signaling | Cell cycle arrest, Apoptotic signaling pathway, Wnt signaling pathway, DNA repair | DNA repair, Apoptotic signaling pathway, Wnt signaling pathway, Cell proliferation |
| Proposed | DNA repair, Cell cycle arrest, Apoptotic signaling pathway, Wnt signaling pathway, Regulation of cell migration, Immune diseases, Tumor necrosis factor signaling, Cell proliferation | DNA damage response, Apoptotic signaling pathway, Wnt signaling pathway, Tumor necrosis factor signaling, Cell proliferation, DNA damage checkpoint signaling | Cell cycle arrest, Apoptotic signaling pathway, Wnt signaling pathway, Notch signaling pathway, Tumor necrosis factor signaling, Cell proliferation | DNA repair, Apoptotic signaling pathway, Wnt signaling pathway, Tumor necrosis factor signaling, Cell proliferation |

DNA damage checkpoint signaling in GBM, and the Notch signaling pathway in LGG. They are absent in other method's results, yet evidence has proven their relevance to cancers.

# H  SIMULATED GENE KNOCKOUT EXPERIMENT

## H.1  WORKFLOW

**Step 1:** The simulation begins by ranking all genes based on node degree disparities calculated from the connectivity matrices of the sub-networks. Node degree is quantified as the number of direct connections each gene has to other genes within the network, serving as a measure of its centrality and influence across different cancer subtypes. To derive the connectivity matrices, we analyze the interactions between genes, where each gene is represented as a node and each interaction as an edge. The degree of each node is then computed to identify highly interconnected genes.

**Step 2:** After ranking, we categorize the genes into two sets: a *high-ranking gene set*, which includes genes exhibiting the largest degree disparities (above a defined threshold based on node degree variance), and a *low-ranking gene set*, composed of genes with minimal degree differences (below the same threshold). Using node degree variance as a threshold ensures our classification is statistically grounded. This method isolates genes that play critical roles in the network dynamics.

**Step 3:** Next, we individually simulate the knockout of genes within the high-ranking and low-ranking gene sets. This process involves transforming their expression values to a baseline non-expression level, which is defined as either zero or a predefined low expression value (such as the mean expression level of the lowest 10% of genes). This transformation mimics the functional loss of these genes. For each gene target in the selected sets, we systematically replace its expression value in the patient samples with the baseline non-expression level.

To ensure robustness and statistical validity, we repeat the simulations multiple times, typically running each simulation for a predetermined number of iterations (e.g., 100 or 1000). Each simulation involves the random selection of a subset of genes from the respective gene set. For the random selection, we define the number of genes to be included in each subset based on a fraction of the total genes in the gene set. For instance, we set $p(\text{select})$ to 10%, which means we select 10% of the genes from the high-ranking gene set and 10% from the low-ranking gene set in each iteration. This approach allows us to assess the impact of knocking out varying combinations of genes while maintaining a consistent sample size across runs. The random selection is performed using a uniform sampling technique to ensure that each gene has an equal chance of being included in the knockout simulation for that run. After each knockout simulation, we record the changes in patient distributions regarding the Shift Rate (SR).

## H.2  SHIFT RATE

**Shift Rate**: The shift rate measures the likelihood of sample groups shifting significantly after a set of genes is knocked out. It accounts for the average distance between samples within a patient group before and after the knockout and compares this distance to an adaptive threshold based on the spread (standard deviation) of samples. Let the distance between a sample within a given group before gene knockout, denoted as $x_i^{\text{before}}$, and after gene knockout, denoted as $x_i^{\text{after}}$, be expressed as $\|x_i^{\text{before}} - x_i^{\text{after}}\|$. The spread of samples within the group before knockout in knockout test $j$ is quantified by the standard deviation $\sigma_j$ of their distances to the centroid of the before group. The shift rate (SR) is defined as: $\Delta_{\text{SR}} = \frac{1}{m} \sum_{j=1}^{m} \left( \frac{1}{n} \sum_{i=1}^{n} \|x_i^{\text{before}} - x_i^{\text{after}}\| > k \cdot \sigma_j \right)$ Where $m$ is the total number of knockout tests, $n$ is the number of samples within the group, $\sigma_j$ is the standard deviation of the distances between the samples before knockout and the centroid of the group in knockout test $j$, and $k$ is a scaling factor (e.g., 1.0 or 1.5) used to determine the threshold for considering a shift.

# I  CASE STUDY

## I.1  BREAST INVASIVE CARCINOMA

Breast Invasive Carcinoma, commonly called BRCA, holds a significant position in cancer research due to its prevalence and clinical importance (Sharma et al., 2010). BRCA represents the most common form of breast cancer, accounting for a substantial portion of cancer-related morbidity and mortality worldwide. Moreover, it is a heterogeneous disease with diverse molecular subtypes, each with distinct clinical behaviors and therapeutic responses. This molecular complexity and clinical diversity make it an ideal candidate for investigating gene networks and deciphering the intricacies of cancer biology. Therefore, in cancer studies, BRCA serves as a cornerstone. Insights gained from BRCA research have huge implications for cancer biology and precision oncology, extending beyond breast cancer to other malignancies.

**- BRCA Subtypes.** Within the used BRCA dataset are various molecular subtypes (patient groups). They are identified based on distinct genetic alterations and clinical features. These subtypes include **luminal A, luminal B, HER2-enriched, basal-like, and normal-like subtypes**, each characterized by specific gene expression patterns and clinical behaviors (Orrantia-Borunda et al., 2022). We give a brief overview of these subtypes:

- **Luminal A:** This subtype is characterized by the expression of estrogen receptor (ER) and/or progesterone receptor (PR) and low levels of the HER2 protein. Luminal A tumors typically have a favorable prognosis and are often responsive to hormone-based therapies.

- **Luminal B:** Luminal B tumors also express ER and/or PR but may have higher proliferation markers such as Ki-67 levels (Sobecki et al., 2017). They can be divided into luminal B HER2-positive (ER/PR-positive, HER2-positive) and luminal B HER2-negative (ER/PR-positive, HER2-negative) subtypes. Luminal B tumors generally have a poorer prognosis compared to luminal A tumors.

- **HER2-enriched:** HER2-enriched tumors overexpress the HER2 protein without expressing hormone receptors (ER/PR-negative, HER2-positive). They are typically aggressive and associated with a higher risk of recurrence. Targeted therapies directed against HER2, such as trastuzumab (Herceptin), are often effective in treating HER2-enriched tumors.

- **Basal-like:** Basal-like tumors are characterized by the absence of hormone receptors (ER/PR-negative) and HER2 amplification (HER2-negative). They often display features similar to basal/myoepithelial cells of the mammary gland and are associated with a poor prognosis. Basal-like tumors are frequently referred to as "triple-negative" (Chacón & Costanzo, 2010) breast cancers due to the lack of expression of ER, PR, and HER2.

- **Normal-like:** Normal-like tumors have gene expression profiles resembling normal breast tissue. They are less common and less well-defined than other subtypes, and their clinical significance is not fully understood.

## I.2  EXPERIMENTS AND ANALYSIS PROTOCOLS

We briefly introduce the background and application cases of the experiments and analysis protocols in the biological validation.

**- Gene Expression Distribution Analysis.** This analysis involves examining the distribution of gene expression levels across different experimental conditions or patient groups to visualize the distribution of expression levels for genes. This analysis has been extensively used in cancer research to explore the expression patterns of key oncogenes and tumor suppressor genes across different cancer types and stages. In studies of cancer patients, researchers may compare the expression distributions of oncogenes and tumor suppressor genes between tumor samples and adjacent normal tissue samples. Differences in expression distributions may indicate dysregulation of these genes in cancer. For instance, gene expression distribution analysis was employed to investigate the expression levels of TP53, a well-known tumor suppressor gene, in various cancer types (Olivier et al., 2010). This analysis revealed significant alterations in the distribution of TP53 expression in different cancer cohorts, showing its potential role as a diagnostic or prognostic marker in malignancies.

**- Differential Gene Expression Analysis.** Differential gene expression analysis has been a cornerstone of transcriptomic studies. This analysis compares gene expression levels between different experimental conditions or sample groups to identify significantly upregulated or downregulated genes. Statistical tests such as t-tests or non-parametric tests are commonly used. For example, cancer patients' and healthy controls' gene expression profiles can be compared to identify dysregulated genes in cancer. Genes with significant differences in expression levels may be further investigated as potential biomarkers or therapeutic targets. For example, researchers performed differential gene expression analysis on RNA-seq data from Alzheimer's disease patients and healthy controls (Twine et al., 2011). This analysis identified a panel of differentially expressed genes implicated in neuroinflammation and synaptic dysfunction, showing molecular pathways associated with Alzheimer's disease progression.

## J   ABLATION STUDIES

In this section, we introduce the ablation studies. We designed the ablations and model variants for each module. This is to verify the effectiveness of the proposed method's core concepts across a diverse set of deep structures and training strategies.

Firstly, we executed experiments utilizing various deep generative models to learn sample embeddings in the sample embedding learning module. The experiment comprised the following model variants:

- `GeSubNet-VAE`: It uses basic VAE to learn sample embeddings by performing clustering tasks on patient samples.

- `GeSubNet-VQVAE`: It uses VQ-VAE to learn sample embeddings by performing clustering tasks on patient samples.

- `GeSubNet-GAN`: It incorporates a GAN structure on top of a basic AE. This model performs sample augmentation while performing clustering tasks on patient samples.

Next, in the gene embedding learning module, we conducted experiments using various graph neural network models to learn gene embeddings. The experiment included the following model variants:

- `GeSubNet-GCN`: A variant utilizes GCN to learn gene embeddings through the link prediction task.

- `GeSubNet-GAT`: A variant utilizes GAT to learn gene embeddings through the link prediction task.

Finally, in the ablation study on the gene network inference module, we experimented and included the following model variants:

- `GeSubNet-OneStep`: A variant removes the entire module and substitutes it with a one-step model.

- `GeSubNet-Conca`: Another one-step variant contains an additional neural layer that uses concatenated sample embeddings and gene embeddings for network classification tasks.

**Ablation.**   We conduct three detailed ablation studies to evaluate the impact of each module in `GeSubNet`. More details can be found in Appendix J. Figure 7 presents the results of the three ablation studies across all variant models. For Patient-M, the proposed sample encoder significantly outperforms all other DGM models across the four network inference tasks (BRCA, GBM, LGG, and OV). For instance, the proposed method achieves an average improvement of 32.3%/31.2%/22.1%/32.3% in terms of GED. The Graph-M ablations show that the method using Neo-GNN consistently performs best, while the other GNN models yield comparable results. For Infer-M ablation, `GeSubNet` significantly outperforms the other objective functions, achieving approximately twice the metric values of its counterparts.

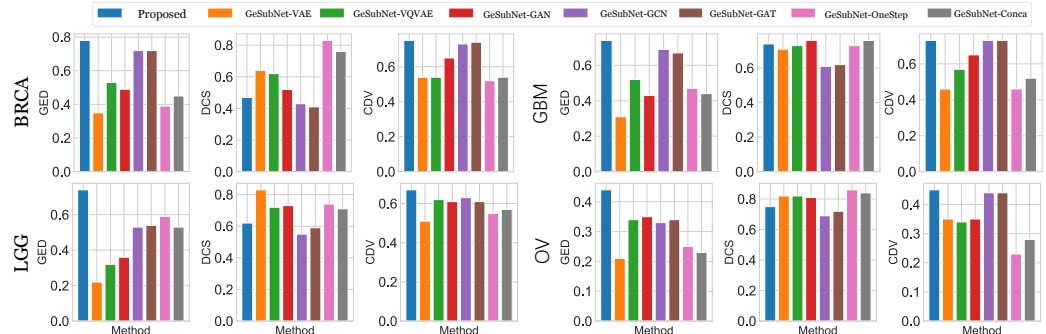

Figure 7: Ablation Study results on GED, DCS, and CDV for the proposed method and all compared methods. GED, DCS, and CDV are subjected to Min-Max normalization.

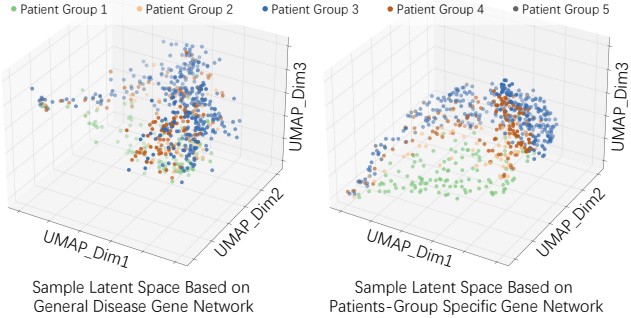

Sample Latent Space Based on
General Disease Gene Network

Sample Latent Space Based on
Patients-Group Specific Gene Network

Figure 8: Compare GNN model learning results based on general and patient group-specific gene networks. The latent sample space was gained via training the GNN model based on the general and patient group-specific gene networks.

## K   PRIOR GRAPH V.S. NEWLY GENERATED GRAPH

We evaluated the performance of patient group learning by inputting the newly generated graph from the `GeSubNet` into a plain GCN and comparing the results. Figure 8 presents a UMAP visualization of the learned latent sample spaces, with the prior graph initialization (Left) and the generated graph GCN initialization (Right). The left sub-figure shows that different patient groups appear mixed in the latent sample space derived from the prior gene network. However, there are clearer boundaries between various patient groups, as shown on the right side. Such results confirm the redundancy of information in the common prior gene networks. It demonstrates that the `GeSubNet` provides more structured information and potential for cancer studies.

