# OpenReview forum: "GeSubNet: Gene Interaction Inference for Disease Subtype Network Generation"
_ICLR.cc/2025/Conference — ICLR 2025 Oral_

### Official Review · Reviewer_X4R1 · 2024-11-02

**Soundness:** 4
**Presentation:** 4
**Contribution:** 3
**Rating:** 8
**Confidence:** 4

**Summary:**

The work presented introduces a novel methodology for gene interaction detection and disease subtype network generation. The topics addressed are of extreme importance in bioinformatics and biomedical machine learning since current methodologies fall short in determining disease subtype networks and limit themselves to the identification of broader networks related to a disease without a fine-grained differentiation between subtypes. The authors present GeSubNet, an architecture composed of three parts. Block 1 is endued with the encoding of gene expression data describing patients. Block 2 deals with the ending of the gene-gene interactions. Block 3 is tasked with determining disease subtype networks. The experiments confirm the biological relevance of the networks obtained and outperform other inference methods also in terms of enrichment analysis, in which GeSubNet is able to detect more enricher terms with respect to its compared counterparts. In the cancer use case presented, the enricher terms are confirmed to be associated with cancer by previous clinical studies.

**Strengths:**

The main strengths of the paper are the following:

1) The work presented is novel. It proposed a new and effective architecture for the task of interest.

2) The methodology is clearly presented, and the experiments are carried out carefully and described thoroughly. In particular, the analysis of the biological relevance of the network further validates the results and highlights the relevance of the methodology for this non-trivial bioinformatics task.

3) The gene knockout simulation is really interesting and seems effective.

4) The method presented outperforms the compared methodologies both quantitatively and qualitatively.

5) Code is provided by the authors allowing for inspection and reproduction of the results.

I believe the points I highlighted make it a good paper. Despite the strong bioinformatics focus, I believe the paper is relevant to the broader machine-learning community, given its substantial methodological contribution.

**Weaknesses:**

Some weaknesses are the following:

1) There are some language inconsistencies in the paper. The authors refer to their method both as GeSubNet and GenSubNet. I believe those are typos.

2) In Figures 1 and 2 the authors seem to have used the "comic sans" font. Please refrain from using that in scientific writing. Moreover, some parts of the text are in bold, italics, or colored. I would suggest writing those parts with no particular font style or color. Also, in lines 103-104, when the authors introduced the name of their methodology, I suggest the authors don't use different fonts to highlight the letters in the words that form the name; bold or capital letters, in this case, work fine.

3) In line 299, introducing abbreviations with a phrase, which is not a clause (so with no verb), sounds not correct. I suggest the authors introduce the abbreviations together with the methods or in the caption of Table 1.

4) There are some misunderstandings about GO terms. The authors refer to "biological functions," but GOs (gene ontologies) are divided into "biological processes," "cellular components," and "molecular function." I guess the authors are referring to "biological processes," but this should be clarified.

Additional typos:

Line 38: "web lab". I guess this should be "wet lab".
Table1: I think that the names that follow #, which stands for "number", should be written in plural.
Line 414: cannot involved -> cannot be involved.

**Questions:**

Overall, I find the paper really interesting. I would simply ask the authors to deal with the "weak points" I described. Moreover, did the authors test the methodology also on different diseases? it would be interesting to see the method applied to detect other disease subtypes. This would further prove the validity and efficacy of the method. However, this would be a "plus" since the work presented, in my opinion, contains enough scientific content for publication.

---

> ### Author Response · Authors · 2024-11-20
>
> We appreciate your detailed review and the recognition of the strength of our work. We have marked the revision in the color **magenta** on the paper.
>
> **W1. There are some language inconsistencies (typos) in the paper.**
> Thank you for pointing out the typos and inconsistencies in the paper.
> We have corrected these issues in the updated version and will have a proofreading after the final decision.
>
> **W2. Issues about the font choices in Figures 1 and 2 and the issues about using text styling (bold, italics, and color) in the manuscript.**
> Thank you for your comments.
> We have revised the figure and text styling in the updated version.
>
> **W3. An issue with how abbreviations are introduced in manuscript line 299.**
> In the updated **line 293**, we have used the abbreviations directly alongside the methods.
>
> **W4. A misunderstanding issue in referring to GO terms as "biological functions".**
> Thank you for pointing out this misunderstanding.
> You are correct that the "biological functions" we referred to are actually "biological processes (BP)" and not "cellular components (CC)" or "molecular functions (MF)." In the revised version, we have clarified in **line 364** that the GO terms we refer to are BP to ensure accuracy. Additionally, we have provided more details about the GO analysis, including the standard GO term categorization (BP, CC, and MF), in Appendix F: GO Function Enrichment Analysis.
>
> **Q1. It would be interesting to see the method applied to detect other disease subtypes.**
> We also want a broader evaluation; however, the limited availability of datasets poses a challenge.
> While our current work focuses on cancer subtype networks, we plan to explore this direction further in future studies. A potential candidate could be the KEGG disease pathway, which presents similar subtyping challenges.

---

> > ### Comment · Reviewer_X4R1 · 2024-11-27
> > **Response to Authors**
> >
> > Thank you for addressing my limited concerns. The paper is good, I keep my score to 8.

---

### Official Review · Reviewer_k9TA · 2024-11-04

**Soundness:** 2
**Presentation:** 3
**Contribution:** 3
**Rating:** 8
**Confidence:** 3

**Summary:**

The manuscript introduces a novel method, GeSubNet, for generating disease subtype-specific networks from general gene-gene interaction networks. The approach combines a deep generative model to learn distinct disease subtypes from patient gene expression profiles and a graph neural network to capture representations of prior gene networks from knowledge databases. By integrating these two representations, GeSubNet generates subtype-specific gene networks. The experimental results demonstrate that GeSubNet outperforms traditional methods across four cancer datasets.

**Strengths:**

1. Generating disease subtype-specific networks from general gene-gene interaction networks is a critical challenge in the field of precision medicine.

2. The manuscript introduces the concept of geometric subnetworks (GeSubNet), with detailed descriptions of the algorithms used and the validation process.

3. The paper is well-written and structured.

**Weaknesses:**

1. The baseline methods used in the comparative analysis are limited. Most of the current baselines are not specifically designed for subtype network generation.

2. The ablation study is limited. A more comprehensive ablation study that evaluates the contributions of individual components (e.g., the
deep generative model and the graph neural network), and the databases, would provide deeper insights into the effectiveness of the model.

3. What are the computational requirements of the GeSubNet algorithm, particularly in terms of runtime and memory usage? How scalable is the method for very large datasets?

4. Figure 1 is hard to understand, particularly the flow of how expression data is integrated into the knowledge graph, as described in Section 3.2.

**Questions:**

see above

---

> ### Author Response · Authors · 2024-11-20
>
> Thank you for your time and for recognizing the strength of this paper. We have marked the revision in the color **blue** on the paper.
>
> **W1. The baseline methods used in the comparative analysis are limited.**
> As this interdisciplinary research represents a newly formulated machine learning task, there are only a few existing works in this area.
> While existing statistical methods may not be explicitly designed for subtype-specific network generation, they calculate pairwise correlation coefficients (e.g., Pearson or Spearman) between gene expression profiles to quantify gene-gene relationships. These coefficients serve as edge weights in the constructed gene networks. In bioinformatics, such methods aim to identify key genes within the network structure or function, aligning with the goals of the proposed method, which generates gene-gene interactions and identifies key genes across subtypes.
> For GNN-based methods, to our knowledge, few studies have focused on this interdisciplinary research. While the baselines are limited and not specifically designed for disease subtype network generation, they have been applied to broader bio-network studies, including disease network generation [1] and bio-function discovery.
> In future work, we will continue to advance this study and incorporate additional related works as they become available.
>
> [1] Murrow L M, Weber R J, Caruso J A, et al. Mapping hormone-regulated cell-cell interaction networks in the human breast at single-cell resolution[J]. Cell systems, 2022, 13(8): 644-664. e8.
>
> **W2. The ablation study is limited.**
> We have included the comprehensive ablation study in Appendix I: Ablation Studies.
> In this section, we evaluated and discussed the contributions of individual modules using four cancer datasets.
> Specifically, we analyzed three variants (GeSubNet-VAE, GeSubNet-VQVAE, and GeSubNet-GAN) based on the deep generative model for Patient-M, two GNN variants (GeSubNet-GCN and GeSubNet-GAT) for Graph-M, and two ablations (GeSubNet-OneStep and GeSubNet-Conca) based on the integration method for Infer-M. Detailed descriptions of these ablations are provided in the appendix.
> In summary, the results demonstrate that the proposed method consistently outperformed the baselines, achieving the best performance.
>
> **W3. What are the computational requirements of the GeSubNet algorithm, particularly in terms of runtime and memory usage? How scalable is the method for very large datasets?**
> We evaluated the proposed method's computational requirements, including runtime (training and inference time) and memory usage, across four cancer datasets. In the updated version of the paper, we have included these analyses mentioned above in a new section titled Appendix E: Computational Requirements.
> Currently, publicly available cancer samples in well-known TCGA typically consist of hundred-level samples. To assess the method's scalability, we conducted an additional test using the TCGA Pan-Cancer expression dataset based on the BRCA graph. This dataset includes 8,314 patient samples and is one of the largest pan-cancer datasets available.
> Overall, the proposed method's computational requirements are manageable for real-world cancer datasets and scale effectively to larger datasets with increased patient samples.
>
> | Dataset | Training Time (sec)     | Inference Time (sec)     | GPU Memory Usage (MB)     |
> |---------|-------------------------|--------------------------|---------------------------|
> | BRCA          | 6.23 ± 0.03              | 2.21 ± 0.05                | 4393 ± 152                |
> | GBM           | 3.64 ± 0.04             |  1.44 ± 0.07              | 2764 ± 117                |
> | LGG           | 5.96 ± 0.03               |    1.98 ± 0.04            | 3834 ± 122                |
> | OV            | 3.45 ± 0.04             |   1.25 ± 0.04             | 2583 ± 108                |
> | Pan-cancer    | 454.47 ± 12.2              |    142.86 ± 5.2          | 15893 ± 733               |
>
> **W4. Figure 1 is hard to understand, particularly the flow of how expression data is integrated into the knowledge graph.**
> We assume you are referring to Figure 2, the system overview.
> We have revised this figure by adding the symbols and loss function equations (Eq. 1-3) mentioned in Sec. 3.2 to both the figure and its caption.

---

> > ### Comment · Reviewer_k9TA · 2024-11-23
> >
> > Thank you for your detailed response. The responses addressed my concern (mainly in W1), so I raised my score to 8.

---

### Official Review · Reviewer_ezgh · 2024-11-04

**Soundness:** 4
**Presentation:** 4
**Contribution:** 4
**Rating:** 8
**Confidence:** 4

**Summary:**

The authors present a novel algorithm that identifies gene-to-gene interactions and disease subtypes. It has three components: A disease subtype from gene expression detection, second a gene network deep learning, and thirdly a part that combines both representations. This approach has out perform several SOTA algorithms.

**Strengths:**

The authors present a novel method for creating gene-gene interactions that are aware of disease subtypes. They also present a new combination architecture and a novel method for integrating knowledge graphs and gene expression data, as well as a shared integrated dataset.
The paper is very clear and presents the underlying biological problem concisely. The mathematical formulation of the problem is well presented as well.
The different experiments conducted are very well displayed and designed.

**Weaknesses:**

The authors did not explain the principal way they selected the baseline methods; in other words, why was this method selected and not others? Why did the authors not add or remove methods?
In table two, while the GeneSubNet is significantly better, how can we know if this is an outlier of the training process or a significant improvement? This can be attained by repeated training and presenting the mean with SD or maybe other statistical insight. Specifically, in line 365, by which statistical measure was the importance ascertained?
Line 416: could you cite why you can infer the "functional distinctions..." this might belong to the conclusions.

**Questions:**

Can you expand on why you think your method outperformed the other methods? Maybe such an addition could be an excellent addition to the discussion.

---

> ### Author Response · Authors · 2024-11-20
>
> We appreciate your time and the recognition of the strength of our work. We have marked the revision in the color **orange** on the paper.
>
> **W1. What is the principal way of selecting the baseline methods?**
> In the updated version of the paper, we have included the principles above in Appendix C.1: The Principle of Selecting Baseline Methods.
> As introduced in the Related Work section, existing studies can be broadly categorized into two approaches: statistical methods and deep graph neural network (GNN)-based methods. We collected baselines from both categories.
> For statistical methods, we first selected two classical and widely used techniques in the bio-network computing field: WGCNA and ARACNe. These methods have significantly contributed to bio-network studies over many years.
> We also included more recently proposed methods, i.e., wTO and LEAP.
> Our selection principles were as follows: (1) the method was recently published (and approximately a decade after the above classical methods), (2) it is user-friendly (with outputs compatible with popular analysis packages), and (3) it has been used as a baseline in related bio-network research [1,2].
> For GNN-based methods, we applied the same selection principles and criteria.
> Furthermore, as this interdisciplinary research represents a newly formulated machine learning task, there are only a few existing works in this area.
> Based on these considerations, we selected three methods: GAERF, LR-GNN, and CSGNN.
>
> [1] Davis-Marcisak E F, Deshpande A, et al. From bench to bedside: Single-cell analysis for cancer immunotherapy[J]. Cancer cell, 2021, 39(8): 1062-1080.
> [2] Murrow L M, Weber R J, et al. Mapping hormone-regulated cell-cell interaction networks in the human breast at single-cell resolution[J]. Cell systems, 2022, 13(8): 644-664. e8.
>
> **W2. Why did the authors not add or remove methods?**
> We believe you may be referring to w/o ablation study of the Patient-M, Graph-M, and Infer-M learning modules. Due to space constraints, we have included the ablation study in Appendix I: Ablation Studies.
> In this section, we evaluated and discussed the contributions of individual modules using four cancer datasets. Specifically, we analyzed three variants (GeSubNet-VAE, GeSubNet-VQVAE, and GeSubNet-GAN) based on the deep generative model for Patient-M, two GNN variants (GeSubNet-GCN and GeSubNet-GAT) for Graph-M, and two ablations (GeSubNet-OneStep and GeSubNet-Conca) based on the integration method for Infer-M. Detailed descriptions of these ablations are provided in the appendix.In summary, the results demonstrate that the proposed method consistently outperformed the baselines, achieving the best performance.
>
> **W3. In table 2 how can we know if this is an outlier of the training process or a significant improvement?**
> Thank you for your comment. We have performed repeated training to evaluate the methods regarding means with SD, and statistical tests show that this is a significant improvement. We have updated this result in Table 2.
>
> **W4. In line 365 by which statistical measure was the importance ascertained?**
> First, we used three graph metrics, CDV, GED, and DCS, to measure the structure differences of the generated network.
> Their statistical measures are based on the fundamental properties of the generated network, such as the distinct nodes and edges.
> Then, we used \#EBF (**line 370** in the updated paper) and shift rate ($\Delta_{\text{SR}}$, **line 397**) to measure the biological significance. Here we have included a more comprehensive introduction and definition of these measurements in Appendix E: Evaluation Metrics.
>
> **W5. In line 416, could you cite why you can infer the "functional distinctions"?**
> We added more citations below for the conclusion sentences in line 416 and its related content in Appendix E: Evaluation Metrics. These citations substantiate the idea that functional enrichment studies often use the number and uniqueness of enriched terms to infer the functional distinction or relevance of specific gene sets.
> [1]Ashburner M, Ball C A, et al. Gene ontology: tool for the unification of biology[J]. Nature genetics, 2000, 25(1): 25-29.
> [2] Wu T, Hu E, et al. clusterProfiler 4.0: A universal enrichment tool for interpreting omics data[J]. The innovation, 2021, 2(3).
>
> **Q1. Can you expand on why you think your method outperformed the other methods?**
> This paper investigates a method and a learning loss function that explicitly infers distinct networks tailored to different patient expression profiles. As noted in the Remark (**line 192**), the loss function in existing methods primarily focuses on common network reconstruction, which often includes irrelevant and noisy gene-gene links. This redundancy may diminish biological significance. In contrast, our proposed method emphasizes subtype-specific gene-gene links, effectively filtering out such redundancies and enhancing biological significance.

---

> > ### Comment · Reviewer_ezgh · 2024-11-26
> >
> > Thank you very much for the clarifications you properly addressed my concerns.

---

### Meta-Review · Area_Chair_K4aD · 2024-12-20

**Metareview:**

This paper proposes an algorithm, GeSubNet, for identifying gene-to-gene interactions by integrating gene expression data and knowledge graphs, which is novel and previously unexplored. Most reviewers find the work very interesting and suggest an acceptance. The authors have also well-addressed the issues raised by the reviewers.

**Additional Comments On Reviewer Discussion:**

The idea is considered novel and effective (ezgh), well-written and structured (k9TA), and clearly presented with careful experiments (X4R1). Most reviewers find the work very interesting and suggest an acceptance. The authors have also well-addressed the issues raised by the reviewers.

---

### Decision · Program_Chairs · 2025-01-22

Accept (Oral)